# Japanese Encephalitis Enzootic and Epidemic Risks across Australia

**DOI:** 10.3390/v15020450

**Published:** 2023-02-06

**Authors:** Morgan Furlong, Andrew M. Adamu, Andrew Hoskins, Tanya L. Russell, Bruce Gummow, Maryam Golchin, Roslyn I. Hickson, Paul F. Horwood

**Affiliations:** 1College of Public Health, Medical and Veterinary Sciences, James Cook University, Townsville, QLD 4811, Australia; 2Australian Institute of Tropical Health and Medicine, James Cook University, Townsville, QLD 4811, Australia; 3Commonwealth Scientific Industrial Research Organisation (CSIRO), Townsville, QLD 4811, Australia; 4Australian Institute of Tropical Health and Medicine, James Cook University, Cairns, QLD 4870, Australia

**Keywords:** Japanese encephalitis virus, Australia, *Culex* spp., waterbirds, feral pigs, epidemiological modelling, ecological modelling

## Abstract

Japanese encephalitis virus (JEV) is an arboviral, encephalitogenic, zoonotic flavivirus characterized by its complex epidemiology whose transmission cycle involves reservoir and amplifying hosts, competent vector species and optimal environmental conditions. Although typically endemic in Asia and parts of the Pacific Islands, unprecedented outbreaks in both humans and domestic pigs in southeastern Australia emphasize the virus’ expanding geographical range. To estimate areas at highest risk of JEV transmission in Australia, ecological niche models of vectors and waterbirds, a sample of piggery coordinates and feral pig population density models were combined using mathematical and geospatial mapping techniques. These results highlight that both coastal and inland regions across the continent are estimated to have varying risks of enzootic and/or epidemic JEV transmission. We recommend increased surveillance of waterbirds, feral pigs and mosquito populations in areas where domestic pigs and human populations are present.

## 1. Introduction

Japanese encephalitis (JE) was first detected in 1859 in Japan with over 6000 cases reported; however, it was not until 1935 before the virus was characterized [1,2]. Japanese encephalitis virus (JEV) was thought to be geographically restricted to Northern Australia, but recent outbreaks in four states in southern Australia confirmed the capacity of JEV as an important transboundary disease [3]. From February to November 2022, JEV had been detected and reported in 85 piggeries and numerous feral pigs in Queensland, Victoria, New South Wales, South Australia and the Northern Territory [4]. Over 40 people were reported to have contracted JEV across the same period with 7 deaths reported to date [4]. The Australian government initiated a “One Health” approach in response to the JE outbreaks [5]. The aim of this response was to determine the distribution of JEV, its potential veterinary and anthropogenic risk factors and possible mitigation measures to be instituted [6]. A recent serosurvey indicated that 1 out of 11 people along the New South Wales (NSW) and Victorian (VIC) border showed evidence of exposure to JEV [7]. This prompted the government to scale up JE vaccine uptake in high-risk areas.

JEV is an encephalitogenic zoonotic arbovirus from the family *Flaviviridae* and the genus Flavivirus [8,9] whose complex epidemiological patterns are influenced by environmental, ecological and immunological factors [10]. JEV has five genotypes (I–V) that are associated with specific climatic and environmental conditions [11,12,13]. More than three billion people live within the JE endemic belt globally, with children under 5 years and the elderly considered highly susceptible to the disease [2]. Middle-aged adults in this region have usually acquired lasting immunity due to past exposure to JEV [14,15,16]. JEV is the major cause of encephalitis in Southeast Asia, with an estimated 100,000 cases occurring annually in this region [17]. JEV is mostly considered a rural disease with epizootic spillover that reflects close interactions between humans and amplifying hosts [18]. Urbanization and increased agricultural production such as intensive pig production and rice farming near peri-urban and/or urban settings, further increase the risk of enzootic spillover for human populations [19,20]. Burke and Leake [21] observed that intense JEV circulation occurs independently of clinical manifestations demonstrating the complex fundamentals involved in disease emergence including viremia in amplifying hosts, the presence of anthropophilic vectors and climatic conditions.

Pigs, both domestic and feral, are known to have high rates of infection and play a vital role as amplifying hosts of JEV due to high levels of viremia that can infect mosquitoes [22] while other mammals are considered dead-end hosts due to insufficient viremia [18,23,24,25]. Australia has the world’s largest feral pig population, estimated to be between 4 and 24 million animals [26,27] concentrated in the northern tropics of the country [5]. The fecundity of sows plays a significant role as they generate susceptible populations that can amplify and maintain JEV [28]. A study by Nitatpattana et al. [29] detected JEV in sentinel piglets in Thailand with intermittent viremia which lasted for about 2 weeks. JE in pigs results in decreased sperm production in boars and abortion or stillbirths in sows with negative economic consequences [30]. 

Limited attention is given to wild waterbirds globally despite their roles in JEV epidemiological dynamics and infection ecology [31]. Over 90 bird species can serve as either amplifying or reservoir hosts of JEV [32]. The *Ardeidae* family is highly susceptible to JEV infection [32,33] as they develop high viral titer and are an excellent source of infection for mosquitoes [9,10,34]. Competent mosquito vectors feed on infected waterbirds before spreading the virus to pigs and humans thereby leading to viral emergence in an area [35]. The migration of infected waterbirds through the waterways from northern to southern regions of Australia was associated with increased flooding events connected to the 2022 La Nina weather events [16,35]. This was postulated to be the likely route for the introduction of genotype IV, detected in late 2021 in the Tiwi Islands, and subsequently in southern Australia [36]. 

Several mosquito species are known to be competent JEV vectors; however, vectors in the *Culex sitiens* subgroup are considered the most important due to their zoophilic feeding patterns [6,37,38]. A vector is deemed competent based on its ability to acquire the virus in nature, transmit the infection when feeding on susceptible hosts and be abundant enough to be significant [39]. *Cx. tritaeniorhynchus* is therefore considered the primary vector of JEV in Asia and the Pacific regions due to its high susceptibility, transmission rate and wide distribution, with other *Culex* species considered secondary vectors [3,39,40]. Conversely, Australia’s diverse habitats have different species of *Culex*, where *Cx. annulirostris* is considered the primary vector due to its extensive distribution [6], further supported through ecological niche modeling by Furlong et al. [41].

Despite the growing importance of JE in Australia, there is a dearth of information highlighting possible high-risk areas of viral transmission. Here, we combined the estimated probability of geographic presence of competent vectors and suspected waterbird hosts with estimates of feral pig densities and domestic piggery locations across Australia to highlight areas at greatest risk of JEV transmission. 

## 2. Materials and Methods

Continental scale JEV risk maps were created for mainland Australia using various combinations of geospatial information, in raster files using a 1 km^2^ spatial resolution, relevant to key vectors and reservoir host species suspected of contributing significantly to the virus’ natural transmission cycle. 

### 2.1. Mosquitoes: Potentially Competent JEV Vector Occurrence Estimates in Australia

Raster files representing the probability of vector presence for *Cx. Annulirostris*, *Cx. Quinquefasciatus* and *Cx. Sitiens* were obtained from the ecological niche models (ENMs) estimated by Furlong et al. [41]. ENMs were created using the maximum entropy modelling software, Maxent, version 3.4.4. Presence occurrence data were obtained from VectorMap and the Atlas of Living Australia (ALA) [41]. Climate data were obtained from the Commonwealth Scientific Industrial Research Organisation (CSIRO) Data Access Portal in the form of nine second resolution climate variables with radiative and elevation lapse rate correction for continental Australia [41]. The vector ENMs were combined to form one raster representing the probability of presence of any of the three species, by the standard probabilistic formula of one minus the product of each species not being present (full expression in Appendix A, in Python 3.8 with rasterio 1.2.1, and GeoPandas 0.12.1 to add the borders. This inherently assumes the estimated presence of any species confers the same risk of JEV transmission.

### 2.2. Waterbirds: Occurrence Estimates of Suspected Waterbird Reservoirs in Australia

ENMs for twelve waterbird species belonging to the *Ciconiidae* (storks) and *Ardeidae* families (egrets, bitterns, herons) were obtained from Graham et al. [42]. ENMs were also simulated using Maxent; however, unlike the mosquito ENMs which were presence-only estimates, taxon-specific background data were generated to run presence–absence simulations [42]. Species occurrence data were obtained from the Australian Atlas of Living Australia, the Queensland Museum and CSIRO, and records older than 1950 were excluded [42]. Climate variables were sourced from Worldclim and represent baseline climatic conditions between 1975 to 2005 adjusted to 0.01 degrees resolution [42]. ENMs were combined using the standard probabilistic approach (see Appendix A) in Python 3.8 with rasterio 1.2.1, to form raster files representing the estimated probability of presence for each waterbird subfamily—storks, bitterns, herons and egrets, with the full list in Appendix A. The final raster used in combination maps was created by combining the raster files of each subfamily.

### 2.3. Domestic Pigs: Estimated Risk of JEV Transmission Given Distance from a Domestic Piggery

The coordinates of domestic piggeries in Australia were obtained from publicly available data provided by the Farm Transparency Project https://farmtransparency.org (accessed on 8 April 2022). Data were de-identified, and only piggeries noted as “open” were included in the final analysis. No information about the number of pigs per farm was available. Coordinates were imported into ArcMap as XY data and converted to a raster representing the distance to the nearest piggery for each cell. Distances were subsequentially scaled to be between 0 and 1 following an exponential function, where 1 represents 0 km from a piggery and 0 represents more than 10 km from a piggery. An exponential function was chosen over other distributions as the majority of vectors remain close to their larval habitats as adults, and 10 km was chosen as the cut-off for JEV risk given this is the estimated maximum flight range of *Cx. annulirostris* [43].

### 2.4. Feral Pigs: Occurrence through Population Density Estimates in Australia

Feral pig distribution and abundance maps were obtained from the Australian Department of Agriculture, Fisheries and Forestry (DAFF), which were published as part of the Biosecurity Innovation Project. The raster data were downloaded as a .tif file which represents an informed estimate of the likely distribution and density of feral pigs based on West’s data [26], literature sources and expert opinion. Feral pig density ranged from 0 to 83 pigs per 16 km^2^ grid. The relative density of feral pigs was normalized by dividing all values by the maximum density using the raster calculator tool in ArcMap.

### 2.5. Humans: Estimated Population Density in Australia

Human density estimates were obtained from WorldPop as a .tif raster file and represent the estimated Australian population in 2020 per grid-cell at a resolution of 3 arc seconds [44]. For the purposes of visualizing the distribution, the human population is first linearly normalized by dividing by the maximum value and then normalized using the power-law y = x^γ for the color map, where γ = 0.1.

### 2.6. Synthesis and JEV Risk Map Combinations

Before map layers could be combined, raster files for each animal mentioned above were converted to the same coordinate system (World Geodetic System 1984), cell size (0.0025, 0.0025) and extent (columns = 16,440, rows = 14,298) using the project raster, resample and clip tools in ArcMap, respectively. Maps were combined using the standard laws of probability, assuming independence between species (so probability of A and B is equal to the product of their independent probabilities, see Appendix A). Due to limitations using the ArcMap raster calculator, these “OR” probabilistic calculations were performed in Python 3.8 with rasterio 1.2.1. Values equal to 0 were set to white to match the map background and Australian state borders were obtained from ArcGIS online.

Due to the transmission cycle of JEV, we explored the enzootic and epidemic risks. For the enzootic risk, we combined the probability of the free roaming reservoir and amplifying species, waterbirds OR feral pigs, with (AND) the vector. This takes into account the possibility of JEV already being established in either water birds or feral pigs. Although there is some evidence of direct pig to pig transmission, the vectors are still required to be present to transmit to humans. For the epidemic risk, we combined the free roaming reservoir and amplifying species, waterbirds OR feral pigs, AND mosquitoes AND domestic pigs. This is to account for domestic piggeries being more strongly monitored than the wild animal populations, but presenting a relatively high risk to humans (through mosquitoes) should JEV be present. We considered these enzootic without explicitly showing the human density as it may not be a dominating factor.

### 2.7. Definition of Terms

Enzootic risk: in our study, we refer to enzootic risk as cryptic circulation of Japanese encephalitis virus that is constantly present in both vertebrate (feral pigs or waterbirds) and invertebrate hosts (mosquitoes) which could result in sporadic cases in humans. Epidemic risk: referred to as the presence of JEV in mosquitoes, domestic pigs, waterbirds or feral pigs with spillover events to humans larger than expected.

## 3. Results

A combined presence estimate based on ENMs [41] for three Culex spp. suspected to be competent for JEV transmission is shown in Figure 1. This represents the estimated distribution of *Cx. annulirostris*, *Cx. sitiens* and *Cx. quinquefasciatus* across Australia. We note that there are no areas of zero probability of combined presence (white). All three species considered were estimated to have a low probability of presence in the inland desert regions of the Northern Territory (NT), South Australia (SA) and Western Australia (WA). A high probability of presence was estimated along the northern and eastern coastlines of the continent and some coastal regions of southwestern WA. Flinders, Kangaroo and King Islands and inland regions within the Murray-Darling Basin are also estimated to be high risk for the Culex spp. considered. 

A combined presence estimate based on ENMs of all the twelve waterbird species considered potential JEV reservoirs (see Appendix A) is shown in Figure 2. The estimated probability of presence was low within the parts of the Great Victorian, Gibson and Great Sandy Deserts. Overall, waterbirds are estimated to have a wide distribution across the continent, including notably among inland regions in Queensland (QLD), New South Wales (NSW) and the Northern Territory (NT).

Figure 3a,b show the estimated density of feral pigs, and distance-related risk from a domestic piggery, respectively. Feral pigs appear to have the highest estimated density in the tropical region of northern Australia with a suspected distribution extending along the continent’s eastern coast and regions of WA (Figure 3a). Regarding domestic piggeries, most of the operations are located in southeastern Australia with the majority being located in NSW (Figure 3b).

The estimated enzootic risk of JEV transmission across Australia given the presence of vectors and either feral pigs or waterbirds is highlighted in Figure 4. A high risk is estimated across the northern coast of Australia stretching from Broome, WA to Cape York, QLD and down along the entire eastern coastal region. Additional high-risk areas for suspected enzootic JEV transmission include southern coastal regions of WA, SA and VIC and Flinders, Kangaroo and King Islands. The risk is estimated to be low in most inland regions across the continent excluding those within the Murray-Darling Basin. 

An estimated epidemic risk of JEV transmission in Australia was calculated given the predicted distributions of vectors, domestic piggeries and either waterbirds or feral pigs. High risk areas aligned with piggery locations extend down the eastern coast and eastern inland regions, especially locations along the Great Dividing Range (Figure 5). Areas of Tasmania (TAS) and WA are also predicted to be at risk of epidemic JEV transmission (Figure 5).

## 4. Discussion

Through this study, we have identified key areas that species distribution or density estimates suggest are the highest risk for potential JEV transmission in Australia. This JEV transmission potential is demonstrated through different combinations of reservoir and amplifying species, and potential vectors, reflecting combined presence probabilities, enzootic or epidemic risks.

Our combined presence estimate for the three *Culex* species (Figure 1) predicted a high to medium probability for the presence of any of these potential mosquito vectors around the coastal regions of the north tropics to Australia’s eastern coastal regions of Queensland, New South Wales, Victoria, South Australia and coastal areas of Western Australia. The inland regions surrounding the Murray-Darling Basin, some parts of Tasmania and Kangaroo, King and Flinders Islands were also estimated to have a high probability for the presence of these JEV vectors. These mosquitoes breed in different habitats ranging from shallow vegetative fresh waters for *Cx. annulirostris*, wastewater and effluent ponds for *Cx. quinquefasciatus* and *Cx. gelidus* and flooded salt marshlands or puddles for *Cx. sitiens* [3,6].

Over 80% of the Australian human population are coastal dwellers (Appendix A). These regions include coastal wetlands, mangroves and other habitats that have conducive environments for mosquito breeding and proliferation. The Australian Government’s (Geoscience Australia) report on climate change in 2011 predicted that all major cities along coastal regions will be potentially affected by rising sea levels, higher tides and more frequent storms [45]. Temperature changes and rising sea levels in coastal urbanized areas may cause changes in the peak and duration of JEV transmission periods [46], as many potential JEV vector species are salt tolerant [47]. *Cx. annulirostris* is considered the major JEV vector estimated to be present in approximately 80% of the Australian continent [6]; however, the role of other mosquito species suspected to be competent JEV vectors should be further investigated. There is a need to focus on *Cx. tritaeniorhynchus* and *Cx. gelidus*, two JEV vectors in Asia, that have recently established in northern Australia [6,48]. The transmission efficiency of JEV in other mosquito species such as *Aedes vigilax*, *Ae. notoscriptus* and *Ae. albopictus* is unclear, but their transmission rate has been demonstrated to be lower than *Culex* spp. [49,50]. 

The ability of waterbirds to travel long distances, share the same environment with mosquitoes and their high densities in breeding areas provide a good opportunity for JEV maintenance, amplification and transmission [51]. In our combined ENM of waterbirds (Figure 2), we considered species in the Ardeidae (herons, egrets, bitterns) and Ciconiidae (storks) families. Our model estimated a high probability for the presence of waterbirds along the coastal areas of Australia and inland regions, especially within Queensland, Northern Territory and New South Wales. Australia’s Nankeen night heron (*Nycticorax caledonicus*), plumed egret (*Ardea intermedia plumifera*), little egret (*Egretta garzetta*) and white-faced heron (*Egretta novaehollandiae*) have been shown to play a significant role in Murray Valley encephalitis virus (MVEV) and the Kunjin strain of West Nile virus (WNV_Kunjin_) dissemination and are highly suspected JEV reservoir hosts considering their widely estimated distribution and complex migratory patterns over a large geographical range [3,28]. The 2022 JEV outbreak in piggeries was mainly around the eastern and southeastern boundary of the Murray-Valley Basin which corresponds with the generalized flyway and breeding season distribution identified for these birds [51]. 

Transmission of JEV to local mosquitoes in the Tiwi islands from migratory birds was postulated as the possible route for the introduction of genotype IV into this region [35,36]. In Japan, China, Indonesia and northern Asia, diverse migratory ardeid birds are distributed and thus have been hypothesized to play a significant role in JEV transmission and subsequent geographical spread into new areas [31]. Detection of genotype IV in southern Australia was attributed to the dissemination of JEV by waterbirds through the inland waterways from northern Australia, with subsequent amplification by domestic and feral pigs [3,6,51]; however, the mechanism of introduction remains speculative. Future studies should aim to unravel the reservoir status of other waterbird species and their role in JEV transmission. Additional research suggests Magpie geese (*Anseranas semipalmata*) congregate in large numbers in northern Australia and southern Papua New Guinea along with wandering whistling ducks (*Dendrocygna arcuata*), cormorants and ibises and are believed to be JEV hosts [52]. Satellite tracking of ibises and spoonbills across eastern Australia has shown the movement of these birds through the north and south corridor west of the Great Dividing Range [53] emphasizing the need to include these species in future JEV modelling studies. 

JEV epidemic risk is considered possible when competent mosquitoes are present near piggeries, in conjunction with an overlapping distribution with infected reservoirs such as feral pigs and/or waterbirds. Humans working in piggeries or those living near pigs are at the highest risk of JEV spillover [5]. From our model (Figure 5), southern Queensland, New South Wales, Victoria, South Australia and some parts of Western Australia are at a medium to high risk of JEV spillover into humans. So far, all of the 45 JE cases reported (with 7 fatalities) in Australia were from New South Wales, Victoria, South Australia, Queensland and the Northern Territory. These areas have a high concentration of piggeries, as observed during the 2022 outbreak across four states, which, coupled with Australia’s population around the coastal regions, predisposes these areas to JEV epidemics. At this stage, it is not possible to calculate the epidemic rate associated with human and pig infections of JEV in Australia. However, a recent serosurvey found that a high proportion (nearly 10%) of people living in ‘high-risk’ areas identified in our study had seroconverted to JEV [7]. Further serosurveys throughout Australia and ongoing monitoring of human cases and pig outbreaks of JEV will confirm whether our predictions are supported. 

Although controlling wild waterbirds is not feasible, proper mosquito control in piggeries close to human habitation should be prioritized. A recent study by Yakob et al. [54] estimated that about 3% of the Australian population could be exposed to JEV infection following its recent geographical spread. Prevention of stagnant water and proper waste management of effluent from piggeries should also be prioritized since *Cx. quinquefasciatus*, a secondary competent vector, has an affinity for effluent water. There has been a suggestion on the potential use of *Wolbachia*-based biocontrol strategies for JE using the same principle employed for dengue control [55]. Feral pig control-measures should also be considered in areas where there is high density of pig farms to reduce the risk of JEV transmission into domestic pig populations, where the rapid amplification of JEV zoonotic risk can occur. With Australia’s feral pig population concentrated in the northern tropical part of the country, there is the possibility of pig-to-pig transmission of JEV [56,57]. These regions will likely become hotspots for JEV enzootic circulation, with the possibility for sporadic cases occurring during periods of increased mosquito activity.

Two JE vaccines for humans are registered in Australia and have been used extensively for travel medicine and by the Australian Defence Force. Imojev is administered to people above 9 months as a single dose, while JEspect is administered to those as young as 2 months of age as a double dose [58]. Both vaccines have been demonstrated to provide ~95% protection against JEV. Vaccination of high-risk individuals during an epidemic confers immunity as it is the most effective means of reducing the incidence of JE in humans. However, it does not affect the transmission cycle of JE [32]. The World Health Organization has advocated for vaccination of people, even in countries considered low risk for JEV, provided those areas are conducive for JEV transmission vis-à-vis the presence of reservoirs, ecological conditions that support virus transmission and shared borders with JE endemic countries [59]. 

Conversely, the vaccination of pigs will reduce virus amplification, the infection rate in mosquitoes and subsequent spillover to humans; however, no licensed JEV vaccines are available for pigs in Australia [18]. Previous studies have suggested that vaccination of pigs is inefficient due to the cost of the vaccine and the complexities surrounding maternal antibodies when vaccinating piglets [1,60,61]. One study recommended vaccination of breeding stock 2–3 weeks before the start of mosquito season [62]. Vaccines are also available for horses which display a similar clinical profile to humans when infected with JEV. Equine JEV vaccines are mandatory in some Asian countries such as Hong Kong (China), Malaysia, Japan and Singapore [63]. 

Limitations of this study include the poor understanding of the natural ecology of JEV. This is particularly the case in Australia, where the virus has not historically been endemic and the relative contribution of waterbirds and feral pigs to the enzootic circulation and possible spillover risk to humans is not known. As such, we have used simple combinations of species using probabilistic approaches with no weights to reflect their relative contributions to the transmission cycle or spillover. Further, data availability on known or suspected species contribution to the transmission cycle is incomplete, requiring estimates of presence in our models for most animals.

The distribution, abundance and vector competencies of Australian *Culex* mosquitoes outside of laboratory experiments are not well known. The distribution and abundance of feral pigs and waterbirds in Australia is also not well understood. The current estimate for the population of feral pigs in Australia is 4 million animals, yet until recently the accepted estimate was 24 million pigs, thus highlighting the variability in reported information that could influence model outputs. The data available for domestic piggeries are incomplete, with Australian Pork Limited estimating about 4300 piggeries across Australia [64], but our data were restricted to a little over 2000 piggeries with complete data. An additional limitation is that the models do not account for seasonality or weather events. Vector population dynamics and waterbird migration patterns are highly associated with precipitation and temperature. Our risk estimates, therefore, constitute best estimates, reflecting relative risks for the enzootic cycles and possible epidemics through domestic piggeries based on available information. 

## 5. Conclusions

Following the recent introduction of JEV into mainland Australia, there is a high likelihood that the virus will establish endemic circulation amongst competent reservoirs. A transdisciplinary One Health approach is highly recommended to unravel the complex ecological and epidemiological dynamics of JEV transmission in the country. This study has highlighted the need for the collection of improved data on the distribution of the key animals and vectors involved in the circulation and transmission of JEV in Australia. The models developed in this study can be used to guide future surveillance systems to identify areas predicted to be at higher risk of JEV transmission and subsequently potential spillover to humans.

## Figures and Tables

**Figure 1 viruses-15-00450-f001:**
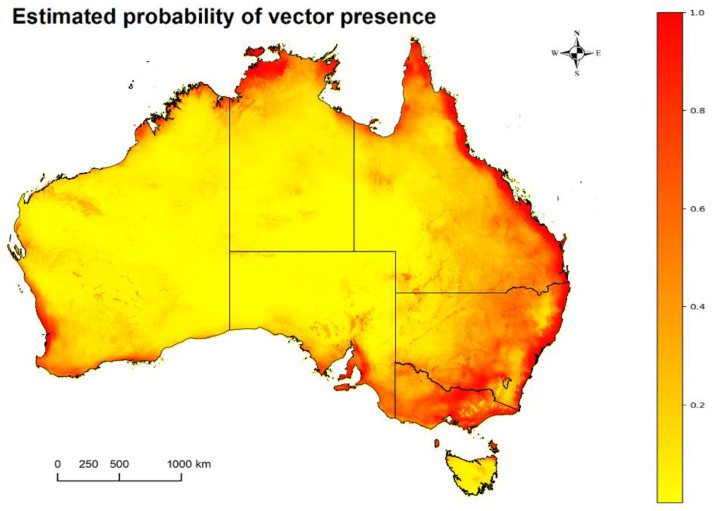
The combined estimated distribution of vectors suspected to be competent for JEV transmission in Australia (*Cx. annulirotris*, *Cx. quinquefasciatus*, *Cx. sitiens*), where the probability of presence ranges from zero (white: none apparent) and low (yellow) to high (red).

**Figure 2 viruses-15-00450-f002:**
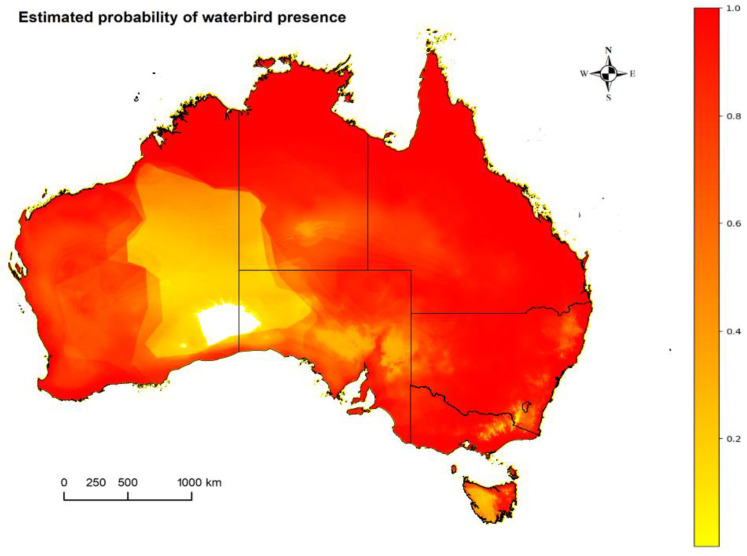
The combined estimated distribution of 12 species of waterbirds suspected to be JEV reservoirs within Australia (from *Ardeidae* and *Ciconiidae* families), where the probability of presence ranges from zero (white) and low (yellow) to high (red).

**Figure 3 viruses-15-00450-f003:**
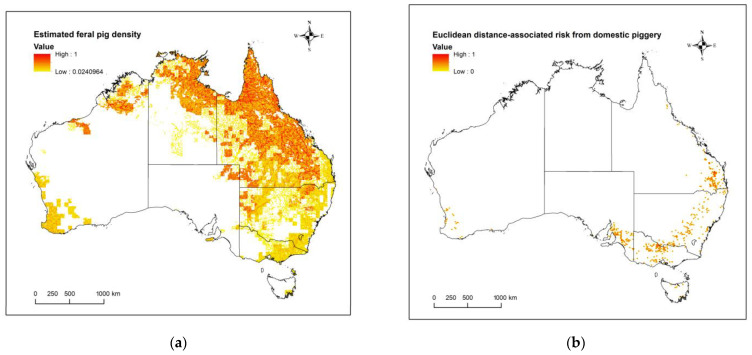
(**a**) represents the relative estimated density of feral pigs in Australia, where the normalized density ranges from 0 (white) and low density (yellow) to 1 (red) (DAFF, 2022). (**b**) represents an exponential relationship for the Euclidean distance from a domestic piggery in Australia, where high risk (red) is the location of a piggery, which exponentially declines as distance increases with values levelling out at 10 km from the piggery denoting low risk (yellow) and beyond that zero risk (white).

**Figure 4 viruses-15-00450-f004:**
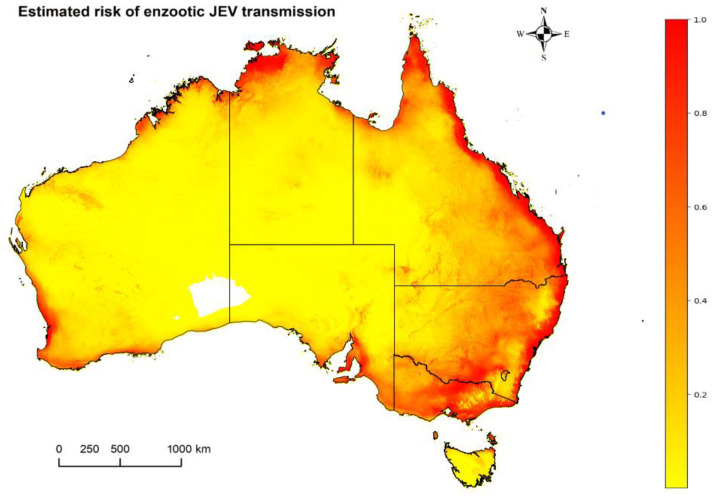
The estimated enzootic risk of JEV transmission in Australia where the estimated risk ranges from zero (white) and low (yellow) to high (red). This shows a combination of free roaming wild species (water birds OR feral pigs) AND the vector presence.

**Figure 5 viruses-15-00450-f005:**
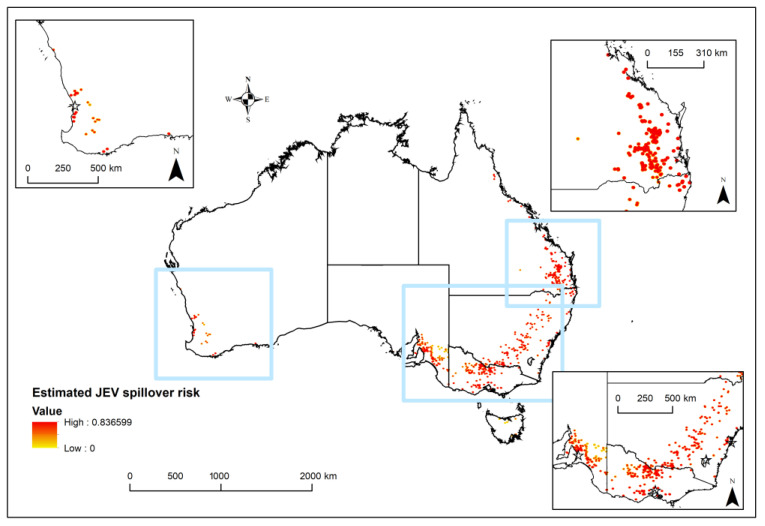
The estimated risk of epidemic JEV transmission in Australia, where the estimated risk ranges from zero (white) and low (yellow) to high (red). This shows a combination of free roaming wild species (water birds OR feral pigs) AND vectors AND domestic pigs. The squares identify the inset regions, expanded for ease of viewing.

## Data Availability

The data presented in this study are available on request from the corresponding author.

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
