# Peer review of "Japanese Encephalitis Enzootic and Epidemic Risks across Australia"

_viruses, 2023, doi:10.3390/v15020450_

Round 1
Reviewer 1 Report
In view of the expansion of epidemic areas of Japanese encephalitis and the unprecedented losses caused in Australia, it is necessary to carry out research to estimate areas at highest risk of JEV transmission in Australia. In this study ecological niche models of vectors and waterbirds, a sample of piggery coordinates and feral pig population density models were combined using mathematical and geospatial mapping techniques, and draw a conclusion that both coastal and inland regions across the continent are estimated to have varying risks of enzootic and/or epidemic JEV transmission.
The conclusion of the article is obtained through calculation and prediction according an indirect /direct data and mathematical model. In order to verify the validity of the conclusion and the reliability of the calculation method, the actual epidemic rate of JE ( both humans and pigs) in high-risk areas (predicted)should be estimated for comparative analysis.
Author Response
Reviewer(s)' Comments to Author:
Reviewer: 1
Comments to the Author
In view of the expansion of epidemic areas of Japanese encephalitis and the unprecedented losses caused in Australia, it is necessary to carry out research to estimate areas at highest risk of JEV transmission in Australia. In this study ecological niche models of vectors and waterbirds, a sample of piggery coordinates and feral pig population density models were combined using mathematical and geospatial mapping techniques, and draw a conclusion that both coastal and inland regions across the continent are estimated to have varying risks of enzootic and/or epidemic JEV transmission.
The conclusion of the article is obtained through calculation and prediction according an indirect /direct data and mathematical model. In order to verify the validity of the conclusion and the reliability of the calculation method, the actual epidemic rate of JE ( both humans and pigs) in high-risk areas(predicted)should be estimated for comparative analysis.
Author Reply
Thank you to the reviewer for this suggestion. As of January 5th, 2023, the Australia’s Department of Health and Age Care has reported 45 JE human cases across states and territories as follows: New South Wales 14, Victoria 14, South Australia 10, Queensland 5 and Northern Territory 2. From the total 45 cases, 7 deaths have occurred - New South Wales 2, Victoria 1, South Australia 2, Queensland 1, and Northern Territory 1. Outbreaks have occurred in 85 piggeries so far in New South Wales, Victoria, South Australia, and Queensland while 55 feral pigs in northern Queensland and Northern Territory tested positive to JEV. The Australian government has not released the distribution of farms associated with outbreaks due to security concerns.
Further text has been added to the manuscript (lines 324-334) to better explain the situation.
Reviewer 2 Report
I really enjoyed to read the paper.
there are several limitations to the study but these are well described in the discussion.
a direct pig to pig transmission is mentioned but no reference added to it.
the conclusion is well described and important.
Author Response
Reviewer(s)' Comments to Author:
Reviewer: 2
Comments to the Author
I really enjoyed to read the paper.
There are several limitations to the study but these are well described in the discussion.
A direct pig to pig transmission is mentioned but no reference added to it.
The conclusion is well described and important.
Author Reply
Thank you to the reviewer for their positive comments.
Two references have been included on line 346 and added to the list of references as listed below.
- Cappelle, J., Duong, V., Pring, L., Kong, L., Yakovleff, M., Prasetyo, D. B., ... & Chevalier, V. (2016). Intensive circulation of Japanese encephalitis virus in peri-urban sentinel pigs near Phnom Penh, Cambodia. PLoS neglected tropical diseases, 10(12), e0005149.
- Ricklin ME, García-Nicolás O, Brechbühl D, Python S, Zumkehr B, Nougairede A, Charrel RN, Posthaus H, Oevermann A, Summerfield A. Vector-free transmission and persistence of Japanese encephalitis virus in pigs. Nat Commun. 2016, 7;10832
Reviewer 3 Report
The article "Japanese encephalitis enzootic and epidemic risks across Australia" applies habitat suitability models for vectors, with distribution of reservoir avian species, domestic and feral pigs. Using this approach they build a series of maps showing high risk areas for transmission of JEV to humans. This work builds on an earlier study focussing on vector distribution. Overall the article is well written with few errors (spelling of tritaeniorhynchus, pg 9), although the Discussion section is overly long. Could the authors also clarify in which figure, human density modelling is applied?
The authors should consider the impact of climate in sub-tropical / temperate regions has on risk as this will not be constant throughout the year. A current concern is that the virus could overwinter and the reports of human infections (ProMed) suggest that it has. Are these cases occurring in areas predicted by their modelling approach?
Author Response
Reviewer(s)' Comments to Author:
Reviewer: 3
Comments to the Author
The article "Japanese encephalitis enzootic and epidemic risks across Australia" applies habitat suitability models for vectors, with distribution of reservoir avian species, domestic and feral pigs. Using this approach they build a series of maps showing high risk areas for transmission of JEV to humans. This work builds on an earlier study focussing on vector distribution. Overall the article is well written with few errors (spelling of tritaeniorhynchus, pg 9), although the Discussion section is overly long. Could the authors also clarify in which figure, human density modelling is applied?
The authors should consider the impact of climate in sub-tropical / temperate regions has on risk as this will not be constant throughout the year. A current concern is that the virus could overwinter and the reports of human infections (ProMed) suggest that it has. Are these cases occurring in areas predicted by their modelling approach?
Author Reply
Thank you to the reviewer for their constructive comments.
We have corrected the spelling of Culex tritaeniorhynchus on Page 9.
Thank you to the reviewer for picking up the discrepancy on the human density modelling. We have added Supplementary Figure S1 to show the human density profile and also updated the Materials and Methods (lines 155-157 and 172-183) to better reflect this part of the study.
We very much agree with the reviewer on the comments about conducting further analysis to consideration seasonal and medium-long term climate change into the model to assess how these factors affect JEV risk. However, we consider these analyses beyond the scope of the current manuscript and they will be included in future research and manuscripts.
All of the human cases that have occurred in Australia so far have been in areas that we have estimated to be at high-risk, or adjacent to these areas. However, due to mobility of humans it is difficult to determine where they were actually exposed to infections. We have updated the discussion (lines 324-334) to address this topic.
Round 2
Reviewer 1 Report
This research is very meaningful. The amount of case data is still slightly insufficient,which might be limited to time or relevant policy relations. I hope that the research team can continue to track and improve the prediction model to make it more accurate.